# Optical Emission Spectroscopy for the Real-Time Identification of Malignant Breast Tissue

**DOI:** 10.3390/diagnostics14030338

**Published:** 2024-02-04

**Authors:** Selin Guergan, Bettina Boeer, Regina Fugunt, Gisela Helms, Carmen Roehm, Anna Solomianik, Alexander Neugebauer, Daniela Nuessle, Mirjam Schuermann, Kristin Brunecker, Ovidiu Jurjut, Karen A. Boehme, Sascha Dammeier, Markus D. Enderle, Sabrina Bettio, Irene Gonzalez-Menendez, Annette Staebler, Sara Y. Brucker, Bernhard Kraemer, Diethelm Wallwiener, Falko Fend, Markus Hahn

**Affiliations:** 1Department of Women’s Health, Tuebingen University Hospital, 72076 Tübingen, Germany; bettina.boeer@med.uni-tuebingen.de (B.B.); regina.fugunt@med.uni-tuebingen.de (R.F.); gisela.helms@med.uni-tuebingen.de (G.H.); carmen.roehm@med.uni-tuebingen.de (C.R.); anna.solomianik@med.uni-tuebingen.de (A.S.); sara.brucker@med.uni-tuebingen.de (S.Y.B.); bernhard.kraemer@med.uni-tuebingen.de (B.K.); diethelm.wallwiener@med.uni-tuebingen.de (D.W.); markus.hahn@med.uni-tuebingen.de (M.H.); 2Erbe Elektromedizin GmbH, Waldhoernlestr. 17, 72072 Tübingen, Germany; alexander.neugebauer@erbe-med.com (A.N.); daninuessle@gmx.de (D.N.); mirjam.schuermann@gmail.com (M.S.); jurjut@tins.ro (O.J.); karen.boehme@erbe-med.com (K.A.B.); sascha.dammeier@erbe-med.com (S.D.); markus.enderle@erbe-med.com (M.D.E.); 3Institute of Pathology and Neuropathology, Tuebingen University Hospital, 72076 Tübingen, Germany; sabrina.bettio@med.uni-tuebingen.de (S.B.); irene.gonzalez-menendez@med.uni-tuebingen.de (I.G.-M.); annette.staebler@med.uni-tuebingen.de (A.S.); falko.fend@med.uni-tuebingen.de (F.F.)

**Keywords:** optical emission spectroscopy, breast cancer, tumor tissue, tumor margin, machine learning, support vector machine, electrosurgery

## Abstract

Breast conserving resection with free margins is the gold standard treatment for early breast cancer recommended by guidelines worldwide. Therefore, reliable discrimination between normal and malignant tissue at the resection margins is essential. In this study, normal and abnormal tissue samples from breast cancer patients were characterized ex vivo by optical emission spectroscopy (OES) based on ionized atoms and molecules generated during electrosurgical treatment. The aim of the study was to determine spectroscopic features which are typical for healthy and neoplastic breast tissue allowing for future real-time tissue differentiation and margin assessment during breast cancer surgery. A total of 972 spectra generated by electrosurgical sparking on normal and abnormal tissue were used for support vector classifier (SVC) training. Specific spectroscopic features were selected for the classification of tissues in the included breast cancer patients. The average classification accuracy for all patients was 96.9%. Normal and abnormal breast tissue could be differentiated with a mean sensitivity of 94.8%, a specificity of 99.0%, a positive predictive value (PPV) of 99.1% and a negative predictive value (NPV) of 96.1%. For 66.6% patients all classifications reached 100%. Based on this convincing data, a future clinical application of OES-based tissue differentiation in breast cancer surgery seems to be feasible.

## 1. Introduction

In 2018, the predicted number of new breast cancers in the European Union (EU) was 404,920, with an estimated age-adjusted annual incidence of breast cancer of 144.9/100,000. Worldwide, there were about 2.1 million newly diagnosed female breast cancer cases in 2018, accounting for almost one in four cancer cases among women [1]. According to GLOBOCAN estimates newly diagnosed breast cancers increased to 2.26 million cases worldwide in 2020 [2]. Also, in the EU, the predicted number of new breast cancer cases in 2020 increased to 530,000 [3]. Less than 1% of all breast cancer diagnoses worldwide affect men [4]. Breast cancer can originate from stem cells or more committed progenitor cells in the mammary ducts or mammary glands [5]. Therefore, early carcinomas are histologically distinguished as being ductal carcinoma in situ (DCIS) and lobular carcinoma in situ (LCIS). Invasive stages are correspondingly named invasive ductal carcinoma (IDC) or invasive lobular carcinoma (ILC). Some invasive breast cancers (IBC) cannot be clearly designated and are, therefore, titled as no special type (NST) [6]. Indeed, the ductal origin dominates and more than 80% of newly diagnosed in situ breast cancers are DCIS [7]. Consequently, ILC accounts for only 5–15% of all invasive breast cancers [8]. Breast-conserving resection with free margins is actually the gold standard for early breast cancer therapy recommended by guidelines worldwide [1,9,10,11,12,13,14]. Re-resection due to positive margins applies to approx. 16–23% of patients [15,16,17]. Yet, repeated surgery affects the recurrence rate, cosmetic outcome, quality of life and cost effectiveness of treatment [18].

Several new approaches for intraoperative tumor margin detection have been proposed during the recent years [19,20]. A new real-time in situ tissue differentiation method is optical emission spectroscopy (OES) [21,22,23]. The tissue differentiation potential of OES during the application of monopolar electrosurgery has already been tested on malignant and healthy human kidney tissues [21,22] as well as individual layers of the human gastric wall [23].

During electrosurgical cutting and the coagulation of a tissue, the high current density at the active electrode leads to ohmic heating. Depending on the waveform, voltage and current settings, cutting or coagulation of the tissue dominates [24,25]. During electrosurgical cutting, water vapor is generated and an explosive expansion of cells as well as extracellular matrix leads to tissue rupture and ejection of tissue fragments [25]. Coagulation denaturates the tissue surface through slower heating. It can take place in direct contact between the electrode and the tissue or in a non-contact way by current sparks [25]. During electrosurgical sparking, a strong electric field is generated leading to the ionization of atoms and molecules in the local tissue vapor and the ignition of a plasma. In this plasma small tissue fragments are further dissociated to molecules and atoms by electron impact [22].

OES enables the quantitative and qualitative analysis of trace elements and electrolytes in tissues after the excitation of individual atoms by energy, e.g., an electrosurgical spark. Excited atoms emit a characteristic electromagnetic radiation (photons with a specific wavelength) which can be captured by a spectrometer with a spectral range of about 200–800 nm [22]. Of note, individual atoms can assume different excitation states depending on the complexity of the atom and the energy input [26].

In this study, normal and abnormal human breast tissue was analyzed regarding differences in the trace element and mineral content by OES during electrosurgical spark formation. Definitionally, abnormal tissue included any form of IBC, tumor necrosis, DCIS, LCIS and tumor stroma. The aim of the study was to determine spectroscopic features which are typical of healthy and neoplastic breast tissue, allowing for future real-time tissue differentiation and margin assessment during breast surgery.

## 2. Materials and Methods

For the ex vivo feasibility study, native human breast tissue from patients > 18 years with invasive breast tumors ≥15 mm was used. All histologic subtypes were included, while patients with neoadjuvant therapies were excluded. The study was approved by the Ethics Committee of the Tuebingen University Hospital, project number 254/2017BO2, and registered at the German Clinical Trials Register (DRKS00012767). The primary aim of the study was to investigate whether abnormal breast tissue yields optical emission spectroscopic features that differ from features of normal breast tissue. Figure 1 summarizes the setup of the study.

### 2.1. Tissue Samples

Fresh tissue samples from breast cancer patients were collected during surgery following informed consent from the patients between April and June 2018. Twenty-four patients with invasive breast cancer >15 mm were enrolled in this study. In accordance with the primary study objective, a pairwise set of normal and abnormal tissue per patient was required. Histopathologic analysis (for details see Section 2.3 Histopathologic evaluation) revealed that both normal and abnormal tissue samples could be obtained from 18 patients (17 females, 1 male; *n* = 4 with invasive breast cancer (IBC) NST, *n* = 11 with IBC NST and DCIS, *n* = 2 with ILC, *n* = 1 with Tubular Carcinoma) (see Table 1). Six patients had to be excluded from further analysis because no representative samples of explicitly histologic normal or abnormal tissue were obtained. No intimate mixtures of normal and neoplastic areas were eligible for machine learning to determine spectroscopic features unique to normal or abnormal breast tissue.

### 2.2. Measurement of Optical Emission Spectra

The experimental setting included a prototype instrument containing an active needle electrode for RF spray coagulation, an optical fiber to collect and transfer light emitted by the RF sparks to a spectrometer unit and a lumen for CO_2_ gas flow to protect the optical fiber from contamination with tissue aerosols and smoke particles. For RF coagulation and generation of sparks, an electrosurgical unit (VIO 3, Erbe Elektromedizin GmbH, Tuebingen, Germany) with the sprayCOAG mode and an effect setting of 2 was used. The CO_2_ gas flow was set to 1 L/min. Optical emission spectra were recorded with a high-resolution Echelle spectrometer (ESA4000, LLA Instruments GmbH, Berlin, Germany) set to an integration time of 100 ms (5 spectra of 20 ms each were summed up to one 100 ms spectra). Potential contamination of the optical fiber by tissue aerosol despite of purge gas flow was monitored by determination of transmission quality using a deuterium/halogen lamp (Ocean Optics Germany GmbH, Ostfildern, Germany) and a Maya 2000Pro spectrometer (Ocean Optics Germany GmbH, Ostfildern, Germany) in regular intervals (after 10 spectra each).

From the 18 patients included, 37 benign and 22 malignant tissue samples were analyzed by generating a total of 992 optical emission spectra. Twenty spectra had to be excluded from analysis because of instrument contamination (*n* = 10), poor spectral quality (*n* = 9) or memory error (*n* = 1), resulting in 972 evaluable spectra (480 from normal, 492 from abnormal tissue) with regard to the primary study objective. Figure 2 provides an overview of the workflow for this study.

### 2.3. Histopathological Evaluation

After OES examination of fresh tissue samples, the precise locations of RF treatment and spectra generation were inked with permanent tissue color for subsequent histological diagnosis and were photodocumented. Subsequently, the tissue samples were fixed in 4.5% neutral buffered formalin, embedded in paraffin and serial sectioning spaced approximately 100–120 µm apart was performed resulting in roughly six sections per sample. All sections were stained with hematoxylin and eosin (HE) stain and evaluated by a pathologist and a second trained observer in a blinded fashion (F.F. and I.G.-M.). For each slide, the ratio of normal tissue versus abnormal tissue was determined, ranging from 100% normal tissue to 100% abnormal tissue. Invasive tumor (IDC, ILC), tumor necrosis, DCIS, LCIS and tumor stroma were considered as abnormal tissue, whereas normal and mastopathic breast parenchyma as well as fatty tissue were defined as normal tissue. For final evaluation, any type of normal or abnormal tissue of any composition (e.g., 80% fatty tissue, 20% parenchyma; or 90% cancerous cells, 10% tumor necrosis) were combined for evaluation of the corresponding recorded spectra. Samples with a mixture of normal and abnormal tissue (e.g., 90% cancerous cells, 10% fatty tissue) were excluded from final analysis, resulting in a total of 59 eligible samples (*n* = 37 normal and *n* = 22 abnormal tissue samples).

### 2.4. Signal Preprocessing and Selection of Spectroscopic Features for Tissue Differentiation

Spectra were first preprocessed with the ESAWin software 16.1.0 (LLA Instruments GmbH, Berlin, Germany). Spectra in which the carbon (C) atom emission line exhibited a low signal-to-noise ratio (<20× standard deviation of the surrounding baseline) were excluded from further analysis. Then, individual emission lines were identified by finding peaks in the spectra and matching them to emission lines of elements occurring in human breast tissue using the LLA Echelle database [27,28]. Eighteen peaks fulfilled this criterion. In addition, the carbon–carbon (C_2_) molecule signal was also considered, as it turned out to be a reliable marker for fat tissue typical of normal breast tissue. The selected peaks had to be higher than 20 times of the standard deviation of the surrounding baseline and had to occur in at least 5% of the spectra to be considered for analysis. Peaks fulfilling these criteria were selected for further analysis.

### 2.5. Classification of Spectroscopic Features and Machine Learning

First, each spectrum was normalized to its total intensity. Then, spectroscopic features were extracted by integrating the signal around the wavelengths of peaks selected in the previous step. The resulting feature vectors are reduced representations of the optical emission spectra, with a focus on a set of emission lines that correspond to chemical elements present in human tissue. Next, feature vectors were split into training and test sets, with the test set containing feature vectors from one patient and the training set vectors from all other patients. We trained a Support Vector Classifier (SVC) on the training set and used it to predict the tissue type (normal vs. abnormal) of feature vectors present in the test set. Prediction correctness was quantified by accuracy, sensitivity and specificity scores as well as positive and negative predictive values (PPV and NPV, respectively). This procedure was repeated for each patient, yielding a different classifier and corresponding classification performance per patient. Data analysis was performed using a custom software written in Python 3.6.8 as well as a set of non-commercial libraries [29].

## 3. Results

### 3.1. Tissue Samples and Patients

According to the histologic assessment, tissue samples from 18 of the originally included 24 patients could be evaluated since they clearly contained normal or abnormal tissue. Table 1 summarizes the characteristics of the 18 patients whose tissue OES spectra were eligible for analysis and machine learning. In addition, it annotates the number of spectra recorded from the normal and abnormal tissue of each patient.

Figure 3 shows examples for (a) a breast tissue sample with coagulation spots derived from RF sparking, and HE stained tissue sections of (b) abnormal tumor tissue, (c) connective tissue with muscular cells and lobular cells and (d) fat tissue with adipocytes.

### 3.2. Selection of Spectroscopic Features

A total of 972 optical emission spectra were used for analysis, comprising 480 and 492 spectra from normal and abnormal tissues, respectively. Optical emission spectra generated by RF sparking on human breast tissue contain bands and emission lines of atoms and molecules derived from the tissue, which includes trace elements and electrolytes. Figure 4 displays six selections of emission lines in detail. Figure 4a depicts emission peaks of phosphor (P) and Zn atoms at 213.618 nm, 213.855 nm and 214.914 nm. The emission line at a wavelength of 247.856 nm (Figure 4b) relates to the excitation of C atoms and the emission line at 248.327 nm is derived from excited iron (Fe) atoms. Emission from Mg atoms can be recorded at 279.522 nm, 279.805 nm, 280.270 nm and 285.213 nm (Figure 4c). Figure 4d shows emission spectra of Ca atoms including peaks at 393.366 nm, 396.847 nm, 422.673 nm and 435.838 nm. The broad peak at 588.995 nm is derived from sodium (Na) atom emission (Figure 4e). Emission lines for potassium (K) (Figure 4f) peak at 766.490 nm and 769.896 nm.

Spectroscopic features to be used in further analysis were detected and identified by a custom designed analysis algorithm. The features were selected according to their occurrence rate in all recorded spectra (more than 5%).

### 3.3. Tissue Classification Per Patient

With all selected spectroscopic features, classification of normal and abnormal tissue was performed by SVC for each individual patient. Figure 5 shows the resulting classification performances with values for accuracy, sensitivity, specificity, PPV and NPV for each individual patient and an arithmetic mean for all patients.

The average classification accuracy of all patients was 96.9%. Normal and abnormal breast tissue could be differentiated with a mean sensitivity and specificity of 94.8% and 99.0%, and a PPV and NPV of 99.1% and 96.1%, respectively (see Figure 5). For 12/18 (66.6%) patients all classifications reached 100%. For patient 10, the accuracy, sensitivity and NPV, at 72%, 44% and 69%, respectively, were exceptionally low. The tumor in patient 10 (NST G3) was relatively small (10 mm) compared to the other tumors, thus the number of spectra generated was restricted (see Table 1). The histology of the tissue from patient 10 revealed 80% of tissue necrosis, which was much higher compared to all other tumors analyzed. Interestingly, the tissue classification for the male (patient 13) showed only minor deviations, with all classifications being higher than 90% and a sensitivity exceeding the average.

Figure 6a shows an example of the emission lines of Mg at a wavelength of 279.552 nm averaged over all normal (green) and abnormal (red) tissue spectra. The excitation of magnesium atoms shows significant intensity differences for normal vs. abnormal breast tissue with a much higher intensity in abnormal tissue. In accordance, the analysis algorithm confirms this spectroscopic feature to be relevant for the discrimination between normal and abnormal tissue, demonstrated by a high feature score. Additional spectroscopic features suitable for tissue discrimination include P at a wavelength of 213.618 nm, Zn at a wavelength of 213.855 nm, C_2_ at a wavelength of 516.49 nm and K at a wavelength of 766.49 nm (Figure 6b). Interestingly, the emission line of Ca at a wavelength of 393.366 nm (Figure 6b) was also comparable between normal and abnormal tissue with only slightly higher intensity in abnormal tissue.

## 4. Discussion

In this study, OES features that can be used to discriminate normal (including healthy or innocuously changed) breast tissue from abnormal (including premalignant or malignant) breast tissue have been established on ex vivo tissue samples from breast cancer patients. Therefore, this study outlines a fundamental step in the clinical application of OES for tumor margin detection in breast cancer.

R0 resection is the primary objective of all curative cancer resections including breast cancer surgery [30]. Guidelines for breast cancer resection define sufficient margins as either “no ink on tumor” which means that the cells on the outer edge of the resectate are no tumor cells, though tumor cells may be found a few cell layers beneath, or recommend about 2 mm distance of neoplastic cells from the resectate surface [1,9,10,11,12,13,14]. Therefore, during breast-conserving surgery, which is attempted for all early-stage breast cancers, close margins are accepted while no tumor tissue should be left in situ. Combined with subsequent radiation therapy, low recurrence rates are observed with the recommended practice [31]. Unfortunately, DCIS often spreads in the ducts and sufficient margins are difficult to achieve by visual estimate [32]. Currently, intraoperative pathological assessment with touch smear and imprint cytology as well as gross tissue inspection combined with specimen radiography is established for margin assessment during surgery [33,34]. These methods have shown high specificity and sensitivity. However, the excellent results come with higher costs since the anesthesia time is prolonged and the operating theater is occupied for, on average, 20–35 additional minutes, decreasing the number of patients who can be treated per day [34]. Moreover, not all health insurances pay for this additional effort, making re-resection inevitable for all patients where residual tumor tissue is found only during subsequent analysis. Re-resection rates in breast-conserving surgery for breast cancer vary being dependent on the subtype, and are about three-times higher for DCIS compared to nonpalpable IBC [15]. Overall, studies reported re-resection rates of about 16–23% which have been slightly decreasing over time [15,16,17]. All patients undergoing re-resection have a delay in adjuvant radio- or chemotherapy. The cosmetic outcome, quality of life as well as cost effectiveness of treatment are compromised [34,35].

Since breast cancer is a common tumor type with about 2 million newly diagnosed cases per year [1], many efforts for an improved margin detection concentrate on this cancer type. Margin probe (radiofrequency spectroscopy), Clear Edge (bioimpedance spectroscopy) and several near infrared spectroscopy (NIRS) devices applicable for indocyanine (ICG)-based tumor detection are already clinically approved [19,20], whereas the first clinical studies have been performed with intelligent knife (iKnife, rapid evaporative ionization mass spectrometry) [36], MassSpecPen (mass spectrometry of water droplets that have been brought into contact with the tissue) [37] and Lumicell (imaging of cathepsin-activatable fluorescent dye) [38]. In addition, breast cancer tissue has been examined by optical coherence tomography (OCT) and diffuse reflectance spectroscopy (DRS) device prototypes [39,40,41,42,43]. Most of the approaches, with the exception of iKnife and NIRS devices, need resected tumor specimens, meaning that too large excisions are not prevented and surgery has to be stopped for analysis. The use of NIRS devices is based on ICG injection. While a complete resection can be easily detected, the method is not directly coupled to the surgical device. The only device apart from OES which can be used during resection is the not-yet-approved iKnife. Though, compared to the spark analysis of OES, iKnife has a delay of up to three seconds until the analysis of the resection site is available, making OES the only real-time margin detection method.

In previous OES studies, human malignant and healthy kidney tissue [21,22] as well as individual layers of the human gastric wall [23] have been analyzed. This study is the first to actually analyze breast tissue and to establish a classification model for normal and abnormal breast tissue.

The sensitivity of 94.8%, specificity of 99.0%, PPV of 99.1% and NPV of 96.1% obtained with OES classification in this study is at least comparable and mostly superior to values obtained with Margin probe (median sensitivity 70–100%, median specificity 70–87%) [44], iKnife (sensitivity 92.1% and 93.4%, specificity 96.4% and 94.9%) [36,45], Clear Edge (sensitivity 84.3% and 87.3%, specificity 81.9% and 75.6%, PPV 67.2% and 63.6%, and NPV 92.2% and 92.4%) [46], MassSpecPen (sensitivity 96.2–100%, specificity 100%) [37], ICG/NIRS (sensitivity 94.2–100%, specificity 31.7–60%) [47,48], DRS combined with autofluorecence analysis (sensitivity 85%, specificity 96%) [49] and OCT (sensitivity 55–80%, specificity 68–87%) [39,41].

In this study, due to training purposes affording clear-cut tissue assignment, six patients with mixed tissue types had to be excluded. Indeed, the OES classification will bring a great benefit for these patients in later clinical use since the abundance of abnormal tissue can be also recognized in mixed tissue areas, which may be difficult to analyze by other methods.

C is very abundant in tissues since it is in the backbone of all biomolecules including fat and proteins [50]. Interestingly, the abundance of C_2_ emission spectra at 516.490 nm was found to be significantly higher in normal tissue compared to abnormal tissue, probably relating to the altered metabolism in tumor cells [51]. Microcalcifications are commonly found in breast tissue [52,53]. Ca hydroxyapatite can be found in benign and malignant tissue, whereas Ca oxalate deposits are only observed in benign breast tissue [53,54]. Mg hydroxyapatite is a typical marker of malignant breast tissue [52,54]. In DCIS, microcalcifications are common though discrimination from benign lesions is challenging [55]. Zn is a cofactor in many different cellular proteins regulating proliferation, homeostasis, immunofunction, oxidative stress, apoptosis and aging [56]. Elevated Zn levels are found in breast tumor tissue and other solid tumors [57,58,59]. Interestingly, Zn levels increase with the grade of malignancy and are different in individual breast cancer subtypes [58]. Tumors have a dysregulated P homeostasis and often overexpress phosphate transporters [60]. Indeed, P is a mitogenic factor and promotes tumor progression. The overexpression of potassium channels is found in many tumors including breast cancer [61]. K channels are involved in the regulation of many cellular processes including cell proliferation and apoptosis. The K homeostasis is frequently changed in tumor cells with both a reduction (hypokalemia) or an increase (hyperkalemia) being observed [62].

Sensitivity, specificity and predictive values for the male breast cancer patient were acceptable, though specificity, PPV and NPV values were below average. This indicates that the OES-based discrimination of breast cancer tissue is probably also suitable in males. Deviations are not surprising since normal and malignant male breast tissue differ from female tissue. Lobes are rare in male breast tissue, and therefore nearly all tumors are of ductal origin. Male breast cancer often exhibits an overexpression of estrogen (ER), progesterone (PR) and/or androgen (AR) receptors, while HER2 is only rarely expressed. Histologically, NST tumors dominate which are often detected as high grade tumors [63,64,65].

Apparently, the high necrosis rate in the tissue samples of one patient (No. 10) interfered with OES-based classification. Necrotic tissue has lost its metabolic function, which is accompanied by an obvious change in trace element abundance and often also structural dissolution as shown in a rat liver tumor model [66]. In tumors, necrotic areas are mostly located in the center which exhibits restricted oxygen and nutrient supply [67]. In addition, local inflammation may trigger tumor necrosis [68].

There are several limitations to our study. The first limitation is the restricted patient number limiting the sample number of different tissue variants. The second limitation is that OES-based tissue classification affords active sparking, respectively, electrosurgical activity on the analyzed tissue preventing scattered tumor islands deeper inside the tissue from being detected. Moreover, contamination of the optical fiber may interfere with spectral quality.

Future studies will investigate the clinical use of OES utilizing an electrosurgical instrument with integrated OES component. In addition, cost-effectiveness can be increased by usage of a spectrometer with reduced resolution.

## 5. Conclusions

The OES technology is a promising solution for intraoperative real-time discrimination between benign and malignant breast tissue with a high sensitivity, specificity, PPV and NPV. A particular strength of this method is the intraoperative independence from other diagnostic means, which prevents longer waiting times leading to a shorter stay of the patient in the operative room. Also, centers without a radiology or pathology department on campus can profit from the benefits of the OES technology. Further research is required for the clinical transfer of OES as a new margin detection method for breast cancer surgery.

## Figures and Tables

**Figure 1 diagnostics-14-00338-f001:**
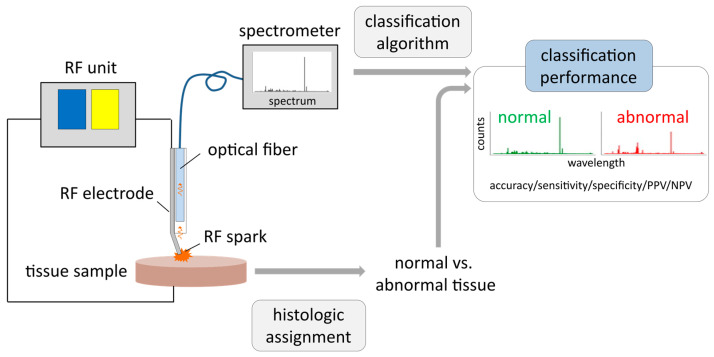
Setup of optical emission spectroscopy (OES) measurements with ex vivo breast tissue samples: Electrical current from a radiofrequency (RF) unit is applied via an active RF needle electrode to a tissue sample. RF sparks excite atoms and molecules derived from tissue fragments. Emitted light is collected via an optical fiber and recorded with a high-resolution optical emission spectrometer. The histologic assignment of tissue samples to either normal or abnormal tissue (mixtures were excluded from analysis) is correlated with the corresponding spectra, evaluating classification performance of normal vs. abnormal breast tissue applying machine learning algorithms. Afterwards, classification performance of included tissue samples is evaluated by accuracy, sensitivity, specificity, positive and negative predictive value (PPV and NPV).

**Figure 2 diagnostics-14-00338-f002:**
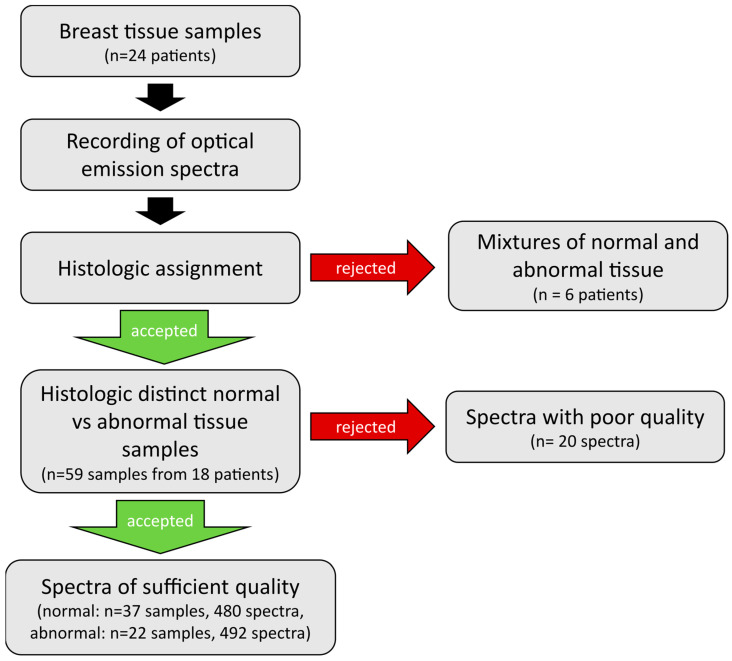
Overview of the workflow of sample and data handling. Optical emission spectra were recorded on breast tissue samples of 24 patients. Histologic assessment of tissue samples revealed mixtures of tissue types for 6 patients, whose tissue samples were excluded from further analysis. The remainder tissue samples of 18 patients were included into analysis, only 20 individual spectra dropped out due to poor quality. Finally, a total of 480 spectra from 37 normal and 492 spectra from 22 abnormal tissue samples were included into evaluation.

**Figure 3 diagnostics-14-00338-f003:**
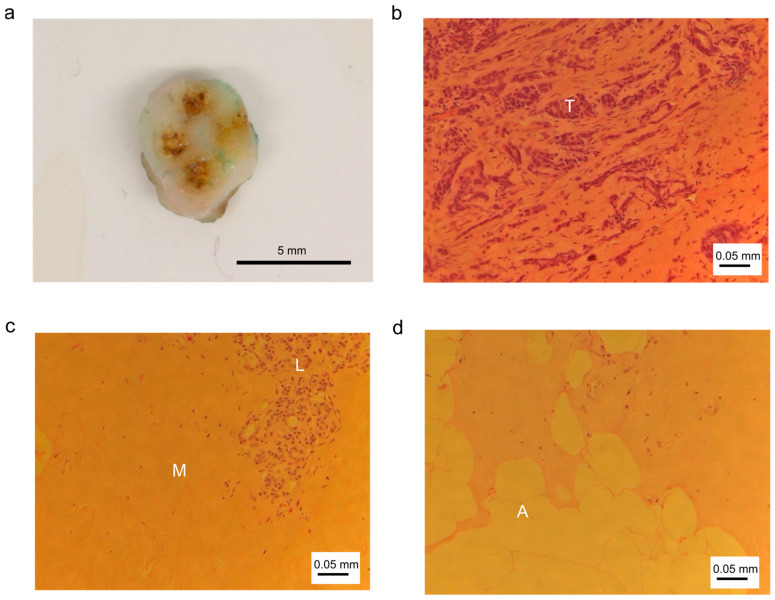
Photographic and histologic images of breast tissue and cells. (**a**) Tissue sample with coagulation marks; (**b**) abnormal tissue: tumor tissue with tumor cells (T); (**c**) normal tissue: connective tissue with muscle cells (M) and lobular cells (L); (**d**) normal tissue: fat tissue with adipocytes (A).

**Figure 4 diagnostics-14-00338-f004:**
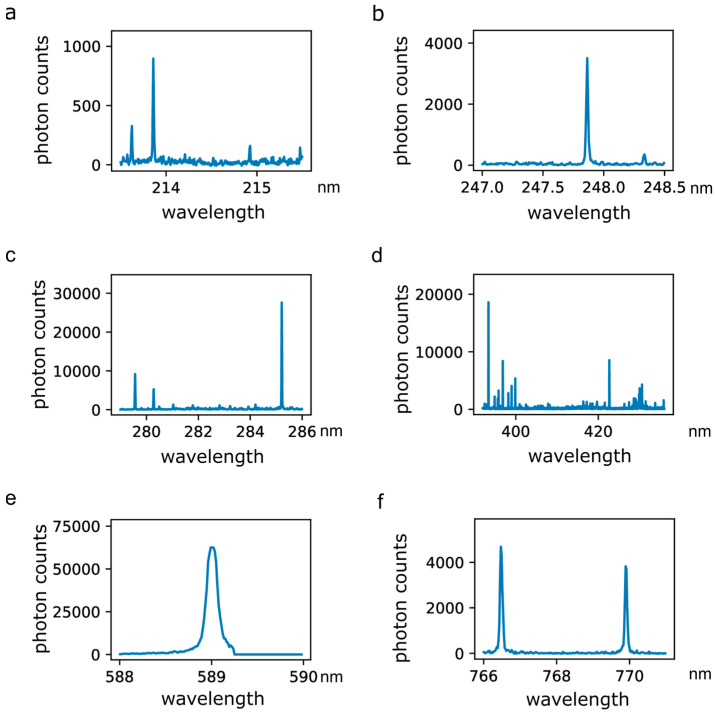
Excerpts of optical emission spectra recorded during the application of RF sparking on human breast tissue. (**a**) Emission spectra of phosphor (P) and zinc (Zn) atoms including peaks at 213.618 nm, 213.855 nm and 214.914 nm; (**b**) emission spectra of carbon (C) and iron (Fe) atoms including peaks at 247.856 nm and 248.327 nm; (**c**) emission spectra of magnesium (Mg) atoms including peaks at 279.522 nm, 279.805 nm, 280.270 nm and 285.213 nm; (**d**) emission spectra of calcium (Ca) atoms including peaks at 393.366 nm, 396.847 nm, 422.673 nm and 435.838 nm; (**e**) emission spectrum of sodium (Na) including a peak at 588.995 nm; (**f**) emission spectra of potassium (K) including peaks at 766.490 nm and 769.896 nm.

**Figure 5 diagnostics-14-00338-f005:**
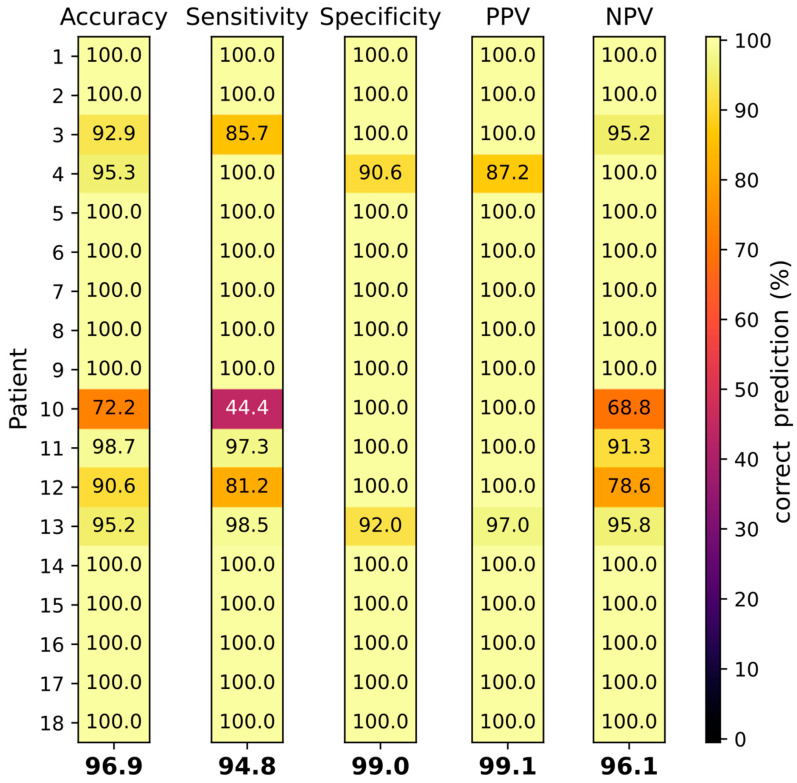
Tissue classification by SVC of 18 individual patients (rows 1–18). Accuracy, sensitivity, specificity, PPV and NPV have been determined (columns from left to right). Correct prediction is depicted by a color range. Values at the bottom of the columns display arithmetic means of all patients.

**Figure 6 diagnostics-14-00338-f006:**
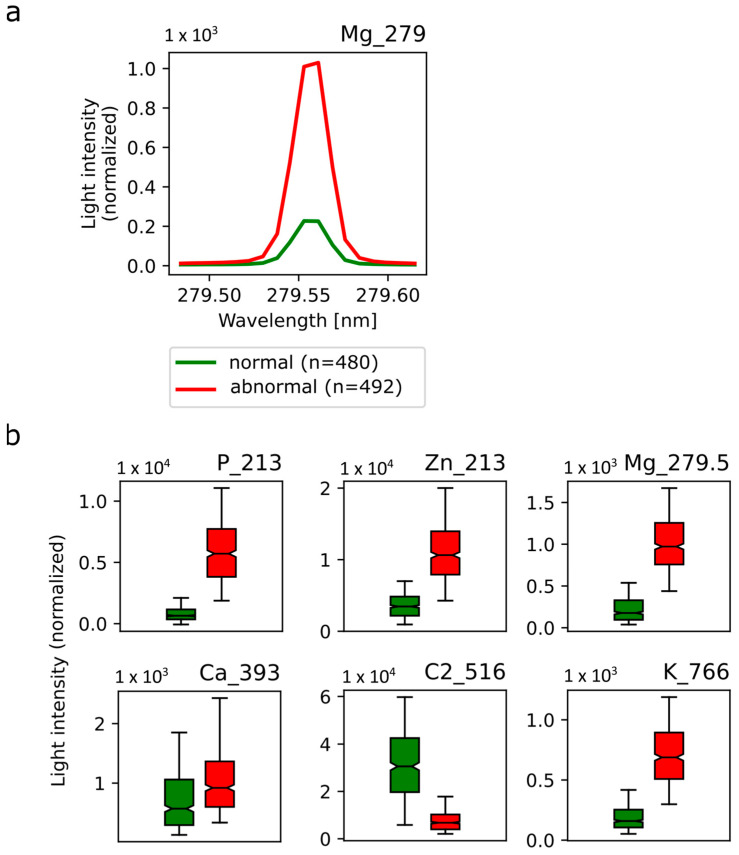
Selected emission spectra in normal and abnormal breast tissue. (**a**) Mean emission line from magnesium 279.552 nm atom excitation in normal breast tissue (green) and abnormal breast tissue (red). (**b**) Mean light intensity was averaged over all normal (green) and abnormal (red) spectra of P, Zn, Mg, Ca, C2 and K at the respective wavelength and depicted in box plots. Corresponding standard deviations are shown.

**Table 1 diagnostics-14-00338-t001:** Patient details including age, sex, histologic diagnosis and number of spectra evaluated.

Patient No.	Age	Sex	Histologic Diagnosis	Number of Spectra Normal Tissue	Number of Spectra Abnormal Tissue
1	77	f	NST G2, DCIS DIN 2	37	39
2	65	f	NST G2, DCIS DIN 2	24	17
3	50	f	Tubular Carcinoma G1	20	7
4	25	f	NST G2, DCIS	64	42
5	77	f	NST G3, DCIS DIN 2	56	3
6	71	f	NST G3, DCIS DIN 2	70	60
7	43	f	NST G2, DCIS DIN 2	14	2
8	55	f	NST G2, DCIS DIN 3 (right breast), ILC G2, LCIS (left breast)	22	15
9	59	f	NST G2, DCIS DIN 2	17	59
10	68	f	NST G3	11	10
11	84	f	NST G3, DCIS DIN 3	21	75
12	77	f	NST G2, DCIS DIN 2	11	16
13	75	m	NST G3	25	65
14	73	f	ILC G2	12	6
15	80	f	NST G2	11	2
16	33	f	NST G2, DCIS DIN 2	9	23
17	63	f	NST G3	27	36
18	85	f	ILC G2	30	17

NST: No special type (ductal or lobular origin unclear); DCIS: ductal carcinoma in situ; DIN: ductal intraepithelial neoplasia; LCIS: lobular carcinoma in situ; ILC: invasive lobular carcinoma; G1–3: Gleason scores (G1 = well differentiated; G3 = poor differentiated); f: female; m: male.

## Data Availability

The data presented in this study are available on request from the corresponding author.

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
