# Peer review of "Optical Emission Spectroscopy for the Real-Time Identification of Malignant Breast Tissue"

_diagnostics, 2024, doi:10.3390/diagnostics14030338_

Round 1

Reviewer 1 Report

Comments and Suggestions for Authors

Dear Authors,

thanks for submitting this interesting study.

The argument is an hot topic, with a relevant impact on clinical practice.

The introduction is exhaustive, the materials and methods are clear, and the discussion is “on point”.

I have just one comment. I suggest specifying from the first mention what is the meaning of “normal and non-normal tissue”.

Author Response

Reviewer 1

Comment 1: Dear Authors, thanks for submitting this interesting study.

The argument is an hot topic, with a relevant impact on clinical practice.

The introduction is exhaustive, the materials and methods are clear, and the discussion is “on point”.

Response 1: Thank you very much for your commending response.

Comment 2: I have just one comment. I suggest specifying from the first mention what is the meaning of “normal and non-normal tissue”.

Response 2: The classification of "normal and non-normal" tissue has been brought up by another reviewer. We changed the phrase “non-normal” to “abnormal” in the text, figure 1, 2 and 6, table 1 as well as the graphical abstract and give a better explanation at the first mention of “normal and abnormal tissue”:

Action 2: (Line 84) Definitionally, abnormal tissue included any form of IBC, tumor necrosis, DCIS, LCIS, and tumor stroma.

Reviewer 2 Report

Comments and Suggestions for Authors

This is a very interesting manuscript which evaluated spectroscopic features of neoplastic and non- neoplastic breast tissue.

As normal and non- normal breast tissue could be differentiated with high sensitivity, specificity, negative and positive predicted values, optical emission spectroscopy clinical application in breast cancer surgery seems to be feasible.

I have no suggestions.

Thank you very much.

Author Response

Reviewer 2

Comment 1: This is a very interesting manuscript which evaluated spectroscopic features of neoplastic and non- neoplastic breast tissue.

As normal and non- normal breast tissue could be differentiated with high sensitivity, specificity, negative and positive predicted values, optical emission spectroscopy clinical application in breast cancer surgery seems to be feasible.

I have no suggestions.

Thank you very much.

Response 1: Thank you very much for the favorable evaluation of our manuscript.

Reviewer 3 Report

Comments and Suggestions for Authors

1.     (Line 41) The authors should consider updating their information, as the data they reference, such as the predicted number of new breast cancer cases in the European Union (EU) from 2018, is almost six years old. Providing more recent statistics would enhance the relevance and accuracy of their study

2.     (Line 122) The sentence has some clarity issues and could be improved for better readability. “Six patients had to be excluded from further analysis because no representative samples of explicitly histologic normal or non-normal tissue were obtained, usually due to the intimate mixture of normal and neoplastic areas. These samples were not applicable to machine learning to determine spectroscopic features unique for normal or non-normal breast tissue.

3.     (Line 47-48): The classification provided indicates that early carcinomas are histologically distinguished as ductal carcinoma in situ (DCIS) and lobular carcinoma in situ (LCIS), while invasive stages are correspondingly named invasive ductal carcinoma (IDC) or invasive lobular carcinoma (ILC). However, on line 111, the authors used n=4 with invasive breast cancer (IBC) NST, n=11 with IBC NST and DCIS, and n=2 with ILC. It's important to ensure consistency in the terminology and classification used throughout the study.

4.     General comment: In medical terminology, the use of the term "abnormal tissue" is more common and widely accepted than "non-normal tissue." "Abnormal" is often used to describe tissue, conditions, or test results that deviate from the normal or expected state. Therefore, it would be better to use "abnormal" instead of "non-normal" for clarity and to align with standard medical terminology.

5.     Line 139 indicates that the authors initially assigned '972 evaluable spectra (480 from normal, 492 from non-normal tissue)' in relation to the primary study objective. However, on line 165, the authors reverted to referring to 59 samples. It would enhance consistency and clarity if the authors continued to use the previously assigned numbers throughout the manuscript.

Comments on the Quality of English Language

Overall, this study presents intriguing findings in the context of optical emission spectroscopy for differentiating breast tissue during surgery. The results demonstrate promise for clinical applications and represent a significant contribution to the field of breast cancer research. However, there are areas for improvement, such as updating outdated data and enhancing the clarity of certain sections. Once these revisions are made, this study has the potential to make a valuable addition to the literature and can be considered for publication.

Author Response

Reviewer 3

Comment 1: (Line 41) The authors should consider updating their information, as the data they reference, such as the predicted number of new breast cancer cases in the European Union (EU) from 2018, is almost six years old. Providing more recent statistics would enhance the relevance and accuracy of their study.

Response 1: Thank you very much for your suggestions. We included more recent literature referring to data of 2020. As 2 new references were inserted all subsequent reference numbers changed which is not shown in the track changes.

Action 1: According to GLOBOCAN estimates newly diagnosed breast cancers increased to worldwide 2.26 million cases in 2020 [2]. Also in the EU, the predicted number of new breast cancer cases in 2020 raised to 530,000 [3].

Comment 2: (Line 122) The sentence has some clarity issues and could be improved for better readability. “Six patients had to be excluded from further analysis because no representative samples of explicitly histologic normal or non-normal tissue were obtained, usually due to the intimate mixture of normal and neoplastic areas. These samples were not applicable to machine learning to determine spectroscopic features unique for normal or non-normal breast tissue.

Response 2: Thank you for your comment. We rearranged the paragraph to make the statement more clear.

Action 2: Six patients had to be excluded from further analysis because no representative samples of explicitly histologic normal or abnormal tissue were obtained. For machine learning to determine spectroscopic features unique for normal or abnormal breast tissue no intimate mixtures of normal and neoplastic areas were eligible.

Comment 3:  (Line 47-48): The classification provided indicates that early carcinomas are histologically distinguished as ductal carcinoma in situ (DCIS) and lobular carcinoma in situ (LCIS), while invasive stages are correspondingly named invasive ductal carcinoma (IDC) or invasive lobular carcinoma (ILC). However, on line 111, the authors used n=4 with invasive breast cancer (IBC) NST, n=11 with IBC NST and DCIS, and n=2 with ILC. It's important to ensure consistency in the terminology and classification used throughout the study.

Response 3: Thank you for your remark. You might have missed the next sentence in the paragraph, where we explain that some cancers cannot be clearly designated and are therefore titled as no special type (NST). In this case, the invasive character of the tumor is clear (IBC) but it cannot be determined whether its origin is ductal or lobular. Moreover, in some patients the tumor may be partly invasive and partly in situ which leads to a double classification in one patient.

Action 3: We adjusted one sentence, to make it more clear:

Some invasive breast cancers (IBC) cannot be clearly designated and are therefore titled as no special type (NST) [4].

Comment 4: General comment: In medical terminology, the use of the term "abnormal tissue" is more common and widely accepted than "non-normal tissue." "Abnormal" is often used to describe tissue, conditions, or test results that deviate from the normal or expected state. Therefore, it would be better to use "abnormal" instead of "non-normal" for clarity and to align with standard medical terminology.

Response 4: Thank you for your suggestion. We agree that "abnormal" is the better phrase.

Action 4: We changed “non-normal” to “abnormal” throughout the manuscript text, in table 1 and in figures 1, 2 and 6 as well as the graphical abstract.

Comment 5: Line 139 indicates that the authors initially assigned '972 evaluable spectra (480 from normal, 492 from nonnormal tissue)' in relation to the primary study objective.

However, on line 165, the authors reverted to referring to 59 samples. It would enhance consistency and clarity if the authors continued to use the previously assigned numbers throughout the manuscript.

Response 5: Thank you for your comment. As you can see in Figure 2, from 18 patients 59 evaluable tissue samples were collected which were used to generate a total of 972 evaluable spectra. We think, that our numbers are quite clear and refer to individual numbers when describing spectra or samples. At the line you refer to it makes no sense to refer to spectra since the selection took place at the level of tissue samples (see rejection 1 in Figure 2).

Comment 6: Overall, this study presents intriguing findings in the context of optical emission spectroscopy for differentiating breast tissue during surgery. The results demonstrate promise for clinical applications and represent a significant contribution to the field of breast cancer research. However, there are areas for improvement, such as updating outdated data and enhancing the clarity of certain sections. Once these revisions are made, this study has the potential to make a valuable addition to the literature and can be considered for publication.

Response 6: Thank you for this positive summary.